# Harvest Stage Dictates the Nutritive Value of Sorghum Stalk Silage by Shaping Its Fermentation Profile and Microbial Composition

**DOI:** 10.3390/microorganisms13092131

**Published:** 2025-09-11

**Authors:** Xiaoqiang Zhao, Ruiyi Liu, Jing Wang, Yawei Zhang, Shuo Zhang, Wenbin Bai, Qingshan Liu, Yuanqing Zhang

**Affiliations:** 1College of Agricultural Economics and Management, Shanxi Agricultural University, Taiyuan 030801, China; 2College of Animal Science, Shanxi Agricultural University, Taiyuan 030801, China; 3Sorghum Research Institute, Shanxi Agricultural University, Jinzhong 030600, China

**Keywords:** sorghum stalk, forage quality, maturity stages, in vitro rumen fermentation, microbial community

## Abstract

The present experiment aimed to investigate the effects of harvest stages on the fermentation quality and nutritional value of sorghum stalk silage. Sorghum stalks were harvested at the three stages (milk, dough, and ripe), chopped, and ensiled for 60 d. Each treatment had five replicates, and the silages were evaluated for fermentation quality, nutritional composition, in vitro rumen fermentation characteristics, and bacterial community profiles. The results showed that the crude protein, neutral detergent fiber, and acid detergent fiber contents decreased significantly with harvest maturity (*p* < 0.05). Consequently, silage from the ripe stage possessed the highest dry matter, relative feed value, and total digestible nutrients (*p* < 0.05). In vitro rumen fermentation indicated that the ripe stage silage exhibited the greatest gas production, and the lowest concentrations of ruminal ammonia–nitrogen (*p* < 0.05). Microbial analysis revealed a shift from dominant epiphytic Proteobacteria to fermentative Firmicutes post-ensiling, with the ripe stage community co-dominated by *Lactobacillus* and *Leuconostoc*, in contrast to the milk stage’s enrichment with *Klebsiella*. In conclusion, harvesting sorghum at the ripe stage is the optimal strategy as it establishes a beneficial microbial community, resulting in silage with superior nutritional value and rumen fermentation efficiency.

## 1. Introduction

The global population is projected to reach 9.7 billion by 2050 [1]. This increase is expected to drive a higher demand for animal-derived foods. Ruminants play an important role in supplying high-quality animal products all over the world. However, the escalating feed costs and the scarcity of high-quality feed due to global climate change have emerged as pivotal constraints impeding the animal industry’s sustainable development [2,3,4]. Therefore, exploration of cost-effective, high-quality feed as viable alternatives to conventional feedstuffs emerges as a pivotal strategy to fostering sustainable animal production [5]. In addition, to mitigate grain competition between humans and animals, developing fiber-rich feeds unsuitable for human consumption can further optimize ruminants’ inherent advantages in the food production chain [5].

Sorghum is a versatile C4 annual grass crop renowned for its resilience to drought, waterlogging, salinity, and poor soil conditions, making it widely cultivated in tropical, subtropical, and temperate regions worldwide [6,7]. In China, sorghum cultivation spanned 676,510 hectares in 2022, with production amounting to 3.1 million tons according to FAOSTAT (https://www.fao.org/faostat/en/#home, accessed on 20 October 2024). In addition, sorghum stalk possesses high feeding value, being rich in fiber, protein, sugar, and essential amino acids [8]. The potential of sorghum stalk as a cost-effective roughage source has been evaluated in Nellore ram lambs at inclusion levels of 30, 40, 50, and 60%, with no adverse effects on the digestibility of nutrients [9]. However, the application of sorghum straw in ruminant production is still limited since it has a stable structure and hard texture and is rich in lignin, cellulose, and hemicellulose [10]. Therefore, to improve the efficient utilization of sorghum straw, it is necessary to implement effective processing methods to soften the straw’s hardness and simultaneously reduce the cellulose content.

Ensiling is a widely used approach for preserving fresh forage and is predominantly driven by lactic acid bacteria producing lactic acid to rapidly lower pH and inhibit spoilage microorganisms under anaerobic conditions [11,12]. Notably, ensiling can minimize nutrient loss and improve the palatability and digestibility of lignocellulose-rich stalk by partially degrading the structural carbohydrates through the action of microbial enzymes and organic acids [13]. In particular, sorghum has low production costs and high biomass yield [14], making it a promising alternative to corn silage without adverse effects on animal performance [15,16]. It was reported that replacing 50% of corn silage with sorghum silage improved meat quality and growth performance of Boer goats [17]. Previous studies have mainly focused on the nutritional value of sorghum silage at a single harvest period. However, research on the fermentation dynamics and associated shifts in microbial communities, especially in relation to different harvesting stages, remains limited. Given that the maturity stage at harvest profoundly influences the chemical composition, moisture content, and fermentability of sorghum stalk, elucidating these relationships is crucial for optimizing silage quality and utilization efficiency in ruminant production systems. Therefore, the purpose of this study was to systematically evaluate how different harvest stages (milk, dough, and ripe) influence the fermentation characteristics, nutritive value, in vitro rumen fermentation dynamics, and associated bacterial community of sorghum stalk silage, so as to identify the optimal harvest time and provide theoretical support for improving the utilization efficiency of sorghum stalk in ruminant production.

## 2. Materials and Methods

### 2.1. Raw Material Preparation and Experiment Design

Sorghum stalks (Jinnuo No. 3) were harvested at three maturity stages (milk, dough, and ripe) from the Sorghum Research Institute, Shanxi Agricultural University, which is located in Yuci city, China. Jinnuo No. 3 is a registered variety developed by the Sorghum Research Institute of Shanxi Academy of Agricultural Sciences, derived from the cross 10480A × L17R. The site has a typical warm temperate semi-humid continental monsoon climate, with an annual temperature of 9.6 °C and precipitation of 456.8 mm on average. At each harvest stage, duplicate samples were manually collected from five representative quadrats (2 m × 3 m), leaving a stubble height of approximately 10 cm. The sorghum stalk samples were chopped into 2–3 cm after ear removal using a straw cutter (KJ-400, Kunjieyucheng Machinery Equipment Co., Beijing, China). One sample of chopped sorghum stalk was packed into a 5 L silage bucket (26.5 cm height, 18 cm diameter, 11.5 cm opening), compacted to a density of 759.7 ± 10.1 kg/m^3^, sealed with a threaded cap, and ensiled for 60 d at 15–25 °C. Another sorghum stalk sample was collected and was placed into self-sealing bag for further bacteria and V-score evaluations.

### 2.2. In Vitro Rumen Fermentation and Experiment Design

The rumen fluid was collected from six healthy Jinnan beef cattle (450 ± 20 kg) quipped with permanent rumen cannulas before morning feeding. The beef cattle were housed in individual pens equipped with feed troughs and had free access to fresh drinking water. The donor cattle were fed twice daily (08:00 and 15:00 h) with a total mixed ration consisting of 40% whole corn silage, 32% corn grain, 10% soybean meal, 7% wheat bran, 7% jujube powder, 1.6% premix, 0.85% sodium bicarbonate, 0.55% salt, and 1% limestone powder on a dry matter basis. All cattle were maintained on this diet with an adaptation period prior to rumen fluid collection. The rumen fluid was pooled and strained through four layers of cheesecloth into a prewarmed (39 °C) insulated bottle. The in vitro rumen fermentation and rumen buffer solution were prepared according to the method described by Menke and Steingass [18]. The rumen fluid and buffer were mixed at a ratio of 1:2 (*v*/*v*) under continuous flushing with CO_2_ in a bottle maintained in a 39 °C water bath. Sorghum stalk silages from different harvest times were dried for 48 h at 65 °C using an electric blast and constant temperature drying oven (Model DHG-9240, Yiheng Scientific Instrument Co., Ltd., Shanghai, China). Then, the dried sorghum stalk silage was ground to pass through a 40-mesh screen. Glass syringes with a calibrated volume of 100 mL were used as incubation vessels. Each syringe contained 0.200 g sorghum stalk silage from different harvest times and 30 mL rumen fluid–buffer mixture. Three syringes were used as replicates for each sorghum stalk silage at different harvest times, with six syringes without sorghum stalk silage used as the blank. The syringes were incubated in an automatic shaker (Jie Cheng Experimental Apparatus, Shanghai, China) at 39 °C for 72 h. During the incubation, the cumulative gas production (GP) was recorded at 1, 2, 3, 4, 6, 8, 10, 12, 14, 16, 20, 24, 28, 32, 36, 40, 48, 54, 60, and 72 h, respectively.

### 2.3. Nutrient Composition Analysis

The dry matter (DM) of sorghum stalk silages was analyzed according to method 930.15 based on the AOAC [19]. The dry matter loss (DML) of sorghum stalk silages was calculated as the difference between post-ensiling and pre-ensiling dry matter contents. Nitrogen (N) and ether extract (EE) concentrations of sorghum stalk silage were determined using a semi-automatic Kjeldahl apparatus (Kjeltec 8400, Foss Analytics, Hillerød, Denmark) and an ANKOM XT10 extraction system (ANKOM Technology, Macedon, NY, USA), respectively, according to AOAC methods 984.13 and 920.39 [19]. The crude protein (CP) of sorghum stalk silage was calculated as total N × 6.25. The crude ash content of sorghum stalk silage was determined in accordance with AOAC method 942.05 [19]. The neutral detergent fiber (aNDF) was analyzed following the procedure of Van Soest et al. [20] with the inclusion of heat-stable α-amylase and sodium sulfite [21]. The acid detergent fiber (ADF) was determined according to AOAC method 973.18 using an ANKOM A200i Fiber Analyzer (ANKOM Technology, Macedon, NY, USA) [19]. The gross energy of sorghum stalk silage was measured with an oxygen bomb calorimeter (Parr 6300, Parr Instrument Company, Moline, IL, USA). The water-soluble carbohydrate (WSC) content of sorghum stalk silage was measured by the dinitrosalicylic acid (DNS) method [22].

### 2.4. Silage and In Vitro Fermentation Parameters Analysis

At the end of ensiling period, sorghum stalk silage was thoroughly mixed, then 30 g sorghum stalk silage was homogenized with 270 mL distilled water for 1 min using a homogenizer (JS30-230, Supor, Shanghai, China). The homogenate was filtered through four layers of cheesecloth, and the pH of the filtrate was measured immediately with a digital pH meter (pHS-25, Shanghai Precision Scientific Instrument Co., Ltd., Shanghai, China). The filtrate was subsequently passed through a 0.45 μm microporous membrane filter. The ammonia–nitrogen (NH_3_-N) concentration of the filtrate was determined using the method described by Broderick et al. [23]. The organic acids of filter and incubation liquid were analyzed using high-performance liquid chromatography (HPLC) following the method of Zhang et al. [24].

The V-Score system was applied to assess the quality of sorghum stalk silage, in which NH_3_-N, acetic acid (AA), propionic acid (PA), and butyric acid (BA) concentrations were used to calculate a score with a maximum of 100 points. The results were classified into three categories: good (>80 points), fair (60–80 points), and poor (<60 points) [25]. Sensory evaluation of silage quality was performed according to the standards of the German Agricultural Society and classified into four grades: excellent, fair, medium, and spoiled.

Aerobic stability of silage samples was assessed according to the method described by Da Silva et al. [26]. About 1 kg of fresh silage was placed in a double-layer self-sealing bag perforated with holes to allow full exposure to oxygen. Then, a thermocouple electrode with a transmission line was inserted into the geometric center of the silage mass, and the other end was connected to a multi-channel data logger (MDL-1048A, SMOWO, Shanghai Tianhe Automation Instrument Co., Ltd., Shanghai, China) to record silage temperature at 5 min intervals. In addition, five thermocouple electrodes in empty silage bags were positioned at the four corners and center of the laboratory to monitor ambient temperature. Aerobic stability was defined as the time required for the temperature at the geometric center of the silage to rise 2 °C above the ambient temperature.

### 2.5. Bacterial Amplicon Sequencing and Data Processing

Genomic DNA was extracted from sorghum straw silage before and after ensiling at different stages of harvest using the E.Z.N.A. Soil DNA Kit (Omega Bio-tek, Norcross, GA, USA) [27], with five replicates for each sample according to the manufacturer’s protocol. DNA concentration and purity were assessed with a NanoDrop ND-1000 spectrophotometer (Thermo Fisher Scientific, Wilmington, DE, USA), and integrity was verified by 1% agarose gel electrophoresis. The V3–V4 hypervariable region of the bacterial 16S rRNA gene was amplified using the primer pair 338F (5′-ACTCCTACGGGAGGCAGCAG-3′) and 806R (5′-GGACTACHVGGGTWTCTAAT-3′) incorporating sample-specific 7 bp barcodes. PCR products were purified using VAHTS™ DNA Clean Beads (Vazyme, Nanjing, China), quantified by a Quant-iT PicoGreen dsDNA Assay Kit (Invitrogen, Carlsbad, CA, USA), and pooled in equimolar amounts. Paired-end sequencing (2 × 250 bp) was performed on an Illumina NovaSeq 6000 platform (Illumina, San Diego, CA, USA) by Personalbio Technology Co., Ltd. (Shanghai, China) using the NovaSeq 6000 SP Reagent Kit (500 cycles).

Raw reads were demultiplexed using the QIIME2 software package (version 2021.08) [28], and barcode sequences were removed with the cutadapt plugin [29]. Paired-end reads were quality-filtered, denoised, merged, and chimeric sequences were removed using the DADA2 plugin [30], resulting in amplicon sequence variants (ASVs). Taxonomic classification was carried out with the classify-sklearn algorithm against the SILVA 138 SSU database using a pre-trained naive Bayes classifier trained on the same V3–V4 region [31]. Prior to diversity analyses, all samples were rarefied to an equal sequencing depth corresponding to the smallest library size. The alpha diversity indices and beta diversity metrics (unweighted and weighted UniFrac distances) were calculated using the q2-diversity plugin in QIIME2. In addition, negative controls, including extraction blanks and PCR blanks, were processed and sequenced in parallel with the experimental samples. These controls yielded negligible read counts, and no consistent ASVs were detected. Therefore, the low-abundance taxa in our dataset were not attributable to kit or laboratory contaminants.

### 2.6. Calculations and Statistical Analysis

The sorghum stalk silage concentration of non-fiber carbohydrates (NFCs) was calculated as follows [32]:
NFC = (1 − CP − EE − NDF − Ash)

The relative feed value (RFV) and total digestible nutrients (TDNs) of sorghum stalk silage were calculated as follows [33,34]:
RFV = DMI × DDM/1.29
TDN = 878.4 − (0.7 × ADF)
where DMI is the ad libitum intake of dry matter in sorghum stalk silage; DDM is digestible dry matter.

DMI = 120/NDF; DDM = 88.9 − 0.779 × ADF

The cumulative gas production was calculated as follows:
GP = (V − V_0_ − GP_0_)/M
where GP refers to the cumulative instantaneous gas production at time t, mL/g DM; V is the syringe reading the at time t, mL; V_0_ is the syringe reading at time 0, mL; GP_0_ is the average gas production of the blank at time t, mL; M is the sorghum stalk silage DM, g.

The gas production values at different incubation time points were fitted to the model of Ørskov and McDonald [35]:
dp = a + b (1 − e^−ct^)
where dp is gas production (mL) at time t; a is the gas production of the fast degradable feed fraction, mL/g DM; b is the gas production of the slowly degradable feed fraction, mL/g DM; c is the gas production rate of the slowly degradable feed fraction, %/h.

The fermentation parameters, nutritional composition, and feeding value across harvest stages were analyzed using one-way analysis of variance (ANOVA) in SPSS 20.0, followed by Tukey’s test for multiple comparisons, with *p* < 0.05 indicating significant differences. The test of Shapiro–Wilk was used for verifying the normality of the data.

Alpha diversity indices and microbial communities were analyzed using the general linear model (GLM) procedure in SAS 9.3, with the model defined as follows:
Y_ij_ = H_i_ + S_j_(H_i_) + e_ij_
where Y_ij_ represents the observed values at different maturity stages before and after ensiling; H_i_ represents the harvest stage; S_j_(H_i_) indicates the ensiling effect within harvest stages; and e_ij_ represents random error. Tukey’s test was applied for multiple comparisons of treatment means, with *p* < 0.05 indicating significant differences.

## 3. Results

### 3.1. Silage Quality of Sorghum Straw at Different Harvest Period

As shown in Table 1, the DM content (*p* < 0.001) gradually increased with harvesting time, reaching 19.37% and 24.21% at the dough and ripe stages, respectively, both significantly higher than that at the milk stage. The concentration of ash, lactic acid (LA), acetic acid (AA), and the LA/AA ratio did not significantly change among harvest stages. Moreover, Sorghum stalk silage at the ripe stage had a significantly greater concentration of DML (*p* < 0.001), EE (*p* < 0.001), RFV (*p* < 0.001), and TDN (*p* = 0.023) compared with the milk and dough stages, while the TDN value did not differ significantly between the dough and ripe stages. The contents of CP (*p* < 0.001), NDF (*p* < 0.001), ADF (*p* = 0.024), and GE (*p* < 0.001) decreased with advancing harvest maturity, with values highest at the milk stage. The ADF content of the milk stage was similar to that of the dough stage, but both were significantly higher than that of the ripe stage (*p* = 0.024).

### 3.2. Silage Sensory, V-Score and Aerobic Stability Evaluation of Sorghum Stalk Silage

After 60 d of ensiling, no moldy odor or butyric acid smell was detected in sorghum stalk silage from any harvest stage. Sensory evaluation classified all silages as “excellent”, with the dough stage showing the highest total score and the most intact structural preservation (Table 2). According to the V-Score, all harvest stages scored full marks with a quality grade of good (Table 3). As shown in Figure 1, the aerobic stability of sorghum stalk silage increased with delayed harvest, with the aerobic stability at the ripe stage exceeded that of the milk and dough stages by 68.8 h and 14.8 h, respectively.

### 3.3. In Vitro Fermentation Parameters of Sorghum Stalk Silage at Different Harvest Stages

As shown in Figure 2, cumulative GP of sorghum stalk silage followed a similar pattern across harvest stages, increasing rapidly within the first 12 h and plateauing after approximately 40 h. At most time points, GP was greatest in the ripe stage and lowest in the milk stage. Moreover, the GP in the ripe stage exceeded the milk and dough stages by 4.06 mL/0.2 g DM and 1.79 mL/0.2 g DM, respectively, after 72 h of incubation. Table 4 shows the in vitro fermentation parameters of sorghum stalk silage at different harvest stages. The GP (*p* = 0.012) at 72 h differed significantly among harvest stages, with the ripe stage significantly higher than the milk stage (*p* = 0.012). The pH values of the incubation liquid were 6.47, 6.61, and 6.64 for the milk, dough, and ripe stages, respectively, with the dough and ripe stages being significantly lower than the milk stage (*p* < 0.001). The NH_3_-N (*p* < 0.001) concentration of incubation liquid at the milk stage were significantly higher than at the dough and ripe stages. Meanwhile, the AA (*p* = 0.026) and PA (*p* = 0.035) concentrations of incubation liquid at the milk stage were also significantly higher than those at the ripe stage.

### 3.4. Sorghum Straw Silage Microbial Profiles at Different Harvest Times Before and After Ensiling

In total, 2,084,097 high-quality sequences were obtained, with over 69,519 for each sample. The high-quality reads were clustered into 3577 ASVs with an average length of 418 bp. All rarefaction curves of the Shannon index based on the observed ASVs tended to reach a plateau at 25,000 sequences, indicating that the sequencing depth was saturated (Figure 3a). The NMDS analysis indicated clear separation between pre- and post-ensiling sorghum stalks at different harvest stages (PERMANOVA, *p* = 0.001), except for the milk and dough stages before ensiling (Figure 3b). Taxonomic analysis at the phylum level revealed that Proteobacteria and Cyanobacteria were the top two phyla in sorghum stalks at all three maturity stages before ensiling, whereas the dominant phylum shifted from Cyanobacteria to Firmicutes after ensiling (Figure 3c). At the genus level, *Chloroplast* and *Mitochondria* were the top two dominant genera before ensiling at the dough and ripe stages, whereas *Hafnia-Obesumbacterium* and *Chloroplast* predominated at the milk stage. After ensiling, *Lactobacillus* and *Chloroplast* became dominant at the dough and milk stages, and *Lactobacillus* and *Leuconostoc* were predominant at the ripe stage (Figure 3d).

### 3.5. Bacterial Community Composition and Diversity of Sorghum Silage Across Harvest Stages and Ensiling

The alpha diversity of sorghum straw microbiota was significantly influenced by harvest stage and ensiling (Table 5). Ensiling significantly reduced Shannon and Faith PD indices at the dough and ripe stages, with no significant changes observed at the milk stage. Before ensiling, the Shannon and Simpson indices showed no significant differences across harvest stages, whereas Faith PD and Observed features indices were significantly lower at the milk stage than at the dough and ripe stages. After ensiling, Shannon and Simpson indices at the ripe stage were significantly lower than those at the milk and dough stages. Similarly, the Faith PD index at the milk stage was significantly lower than at the dough stage. The Observed features index at the milk stage was consistently lower than those at the dough and ripe stages before ensiling, regardless of ensiling status.

To further investigate the effects of harvest stage and ensiling on sorghum straw microbiota, the GLM was first applied to assess the fluctuations of the top 20 microbiota of sorghum straw under different harvest stages and silage treatments (Table 6). The results indicate that the top 20 microbiota were all significantly influenced by harvest stage and ensiling. LEfSe analysis further identified that the relative abundance of *Chloroplast*, *Sphingomonas*, *Allorhizobium-Neorhizobium-Pararhizobium-Rhizobium*, *Methylobacterium-Methylorubrum, Sphingobacterium*, *Stenotrophomonas*, *Chryseobacterium*, *Devosia*, *Aureimonas*, *Ochrobactrum* and *Massilia* were higher at the dough stage before ensiling. The relative abundance of *Mitochondria*, *Pantoea*, *Hymenobacter*, *Pseudomonas*, and *Flavobacterium* was increased at the ripe stage, and *Hafnia-Obesumbacterium* and *Enterococcus* were higher at the milk stage before ensiling. After ensiling, the relative abundance of *Leuconostoc* was significantly higher at the ripe stage, whereas *Lactobacillus*, *Pediococcus*, *Klebsiella*, and *Weissella* were significantly enriched at the milk stage (Figure 4a,b). Further, Random Forest analysis was employed to screen the most responsive microorganisms to ensiling and harvest stages, with the top 5 taxa identified as *Perlucidibaca*, *Rubellimicrobium*, *Allorhizobium_Neorhizobium_Pararhizobium_Rhizobium*, *Variovorax*, and *Siphonobacter* (Figure 4c). By intersecting key microorganisms identified through the three methods, a total of 18 key bacteria were determined, including *Lactobacillus*, *Leuconostoc*, *Chryseobacterium*, *Pantoea*, *Aureimonas*, *Enterococcus*, *Weissella*, *Chloroplast*, *Sphingomonas*, *Mitochondria*, *Sphingobacterium*, *Hymenobacter*, *Klebsiella*, *Pseudomonas*, *Allorhizobium_Neorhizobium_Pararhizobium_Rhizobium*, *Methylobacterium_Methylorubrum*, *Hafnia_Obesumbacterium, and Pediococcus* (Figure 4d).

### 3.6. Correlations Between Key Bacteria and Fermentation Parameters After Ensiling

The correlations between key bacteria, gas production and fermentation parameters are demonstrated in Figure 5. The results indicated that the concentration of AA was significantly positively correlated with the relative abundance of *Lactobacillus*, Klebsiella and *Aureimonas* and negatively correlated with the relative abundance of *Leuconostoc*. Similarly, the concentration of *isobutyric acid* was also negatively correlated with the relative abundance of *Leuconostoc*. In addition, the concentration of NH_3_-N was significantly negatively correlated with the relative abundance of *Pseudomonas*, *Sphingobacterium*, and *Sphingobacterium* (Figure 5).

## 4. Discussion

DM content is an important factor determining the nutritional value of silage feed, with a higher DM content indicating higher feeding value [36]. In the present experiment, the DM content of sorghum stalk silage gradually increased with harvesting time, and reached the highest level at the ripe stage. This result is consistent with numerous previous studies on forage maturation, in which DM content increases with delayed harvest time [36,37,38], which may be related to the progressive conversion of sugars into starch and the decrease in moisture content during the maturation process [36]. Moreover, the dietary CP content is a critical indicator for determining the nitrogen utilization efficiency (NUE) in ruminants, as lower dietary CP can improve NUE [39], but chronic protein deficiency will also inhibit animal production performance. In addition, the dietary concentrations of NDF and ADF are major factors limiting the growth performance of ruminants, while lower NDF and ADF are associated with greater feed intake and nutrient digestibility [40,41]. In the present experiment, the sorghum stalk silage concentrations of CP, NDF and ADF decreased with advancing harvest maturity, being highest during the milk stage. Consistent with this result, the previous study has indicated that the CP content of forages has been shown to decreases with advancing maturity [42], which may be due to the dilution of protein concentration by the accumulation of structural components. The lower contents of NDF and ADF in ripe stage could be related to the variations in microorganisms during silage. From a practical perspective, the reduced CP concentration at the ripe stage could negatively affect nutrient digestibility, suggesting that additional protein supplementation may be required when formulating rations, in order to balance the diet and sustain optimal animal performance. However, it should be noted that although the CP level declined to 7.81% at the ripe stage, it was still higher than typical values reported for corn stover silage (around 5.0–7.0%) [43,44]. Notably, although the CP content was lower, the RFV and TDN were highest at the ripe stage. RFV and TDN have been widely used to evaluate the feed value of pasture and silage, while higher RFV and TDN values reflect a greater nutritional value of silage [45]. Thus, these results indicate that the ripe stage is the optimal period for sorghum straw silage. However, the pH value was highest at the ripe stage, reaching 4.26. Research has indicated that the lower the pH value, the higher the quality of silage, and the pH value of high-quality silage should not exceed 4.2 [46]. Although the pH of the mature sorghum straw silage was slightly elevated above the ideal value, its quality was confirmed as excellent by both V-score and sensory evaluation, indicating that this minor deviation did not compromise the overall preservation.

Aerobic stability reflects the degree of silage preservation after aerobic exposure and is defined as the time required for the silage temperature to increase by 2 °C above the ambient temperature [47]. Silages with high aerobic stability resist spoilage for a longer time upon exposure to oxygen, thereby preserving their nutritional value and palatability while reducing the risk of mold and mycotoxin contamination [48]. In the present study, the aerobic stability of the sorghum stalk silage increased with advancing harvest maturity, and was highest at the ripe stage and lowest at the milk stage, which may be attributed to the higher moisture content of the milk stage. The higher moisture content resulted in greater effluent production during ensiling and consequently facilitated the proliferation of spoilage microorganisms such as yeasts and other aerobic bacteria.

The in vitro fermentation technique replicates the fermentation dynamics within the rumen, providing a method to assess the degradability of various feedstuffs by rumen microorganisms [48]. The volume of GP is a significant factor for estimating the fermentable nutrient content of a feed [49]. In the present experiment, the highest GP was observed in the ripe stage silage. This result indicates a high content of readily fermentable substrate, which is consistent with the high DM and NFC in ripe sorghum stalk. However, compared to the milk and dough stages, silage from the ripe stage produced lower concentrations of AA, PA and NH_3_-N. Rumen volatile fatty acids (VFAs) are primarily the end-products of cellulose fermentation [50], while NH_3_-N is derived from the degradation of N compounds [51]. Therefore, the decrease in AA and PA at the ripe stage could be attributed to its lower NDF and ADF content, as well as the reduced digestibility of this fiber as it becomes more lignified with maturity; the decrease in NH_3_-N is directly related to its lower crude protein content.

The process of ensiling is fundamentally a microbially driven fermentation, where the composition and dynamics of the microbial community critically determine the final fermentation quality and nutritive value of the silage [52]. In the present experiment, the bacterial composition and diversity of sorghum stalk were significantly reshaped by both the harvest stage and the ensiling process. Specifically, the ensiling shift from Proteobacteria and Cyanobacteria to Firmicutes indicates successful anaerobic fermentation. The Firmicutes phylum encompasses the majority of crucial lactic acid bacteria (LAB), such as *Lactobacillus* and *Leuconostoc*, which are the primary drivers for the rapid acidification necessary forage preservation [53].

Harvest stage proved to be a critical factor in shaping the bacterial community structure and fermentation quality of sorghum stalk silage [54]. The microbial profile at the ripe stage was distinguished by the co-dominance of *Lactobacillus* and *Leuconostoc*. The *Lactobacillus* drives the primary acidification via lactic acid production [55], while the *Leuconostoc* produces both lactic acid and acetic acid [56], which is known to enhance aerobic stability by inhibiting yeasts and molds [57]. Furthermore, the correlation analysis also revealed that the relative abundance of *Lactobacillus* was significantly positively associated with AA, which indicated that the dominance of *Lactobacillus* in the ripe silage primarily contributed to efficient lactic acid fermentation rather than excessive acetate accumulation, consistent with the superior overall fermentation quality observed at this stage. Furthermore, the alpha diversity results support this conclusion, as the lower Shannon and Simpson indices in the ripe stage silage did not indicate a poor ecosystem but rather a highly successful fermentation, where a few elite species (*Lactobacillus* and *Leuconostoc*) outcompeted other microbes, leading to a more stable and efficient preservation. In contrast, the milk stage was significantly enriched with *Klebsiella*, an enterobacterium that competes for substrate and indicates less ideal fermentation [58]. Moreover, a significant positive correlation was also observed between *Klebsiella* and AA, indicating that part of the acetate in the milk stage silage likely originated from *Klebsiella*-driven fermentation, which is associated with ideal fermentation and explained the inferior preservation efficiency at milk stage. Therefore, our findings collectively suggest that harvesting sorghum stalk at the ripe stage is crucial for establishing an optimal microbial community that ensures a more efficient and stable fermentation.

## 5. Conclusions

Harvest maturity critically influenced the fermentation characteristics, microbial community, and nutritional value of sorghum stalk silage, with the ripe stage identified as the optimal harvest time. Silage harvested at the ripe stage exhibited superior nutritional quality, evidenced by the highest relative RFV and TDN, and showed the greatest in vitro rumen fermentability. This enhanced quality was driven by the formation of a unique microbial community co-dominated by *Lactobacillus* and *Leuconostoc*, which contributed to its significantly improved aerobic stability. Nevertheless, the reduced crude protein concentration at this stage suggests that during dietary supplementation with ripe-stage sorghum stalk silage, additional protein supplementation may be necessary to ensure a balanced diet and sustain optimal animal performance. Therefore, harvesting sorghum stalk at the ripe stage is an effective strategy to produce high-quality silage with enhanced nutritional value and robust preservation characteristics.

## Figures and Tables

**Figure 1 microorganisms-13-02131-f001:**
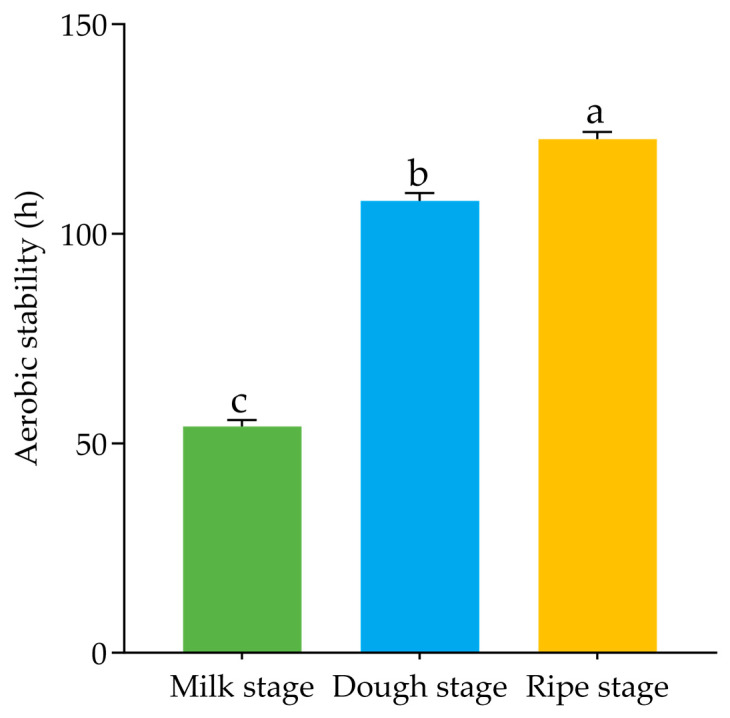
Aerobic stability of different harvest periods of sorghum stalk silage. Different lowercase letters in the figure indicate significant differences among different harvest periods (*p* < 0.05). Aerobic stability was defined as the time required for the temperature at the geometric center of the silage to rise 2 °C above ambient temperature.

**Figure 2 microorganisms-13-02131-f002:**
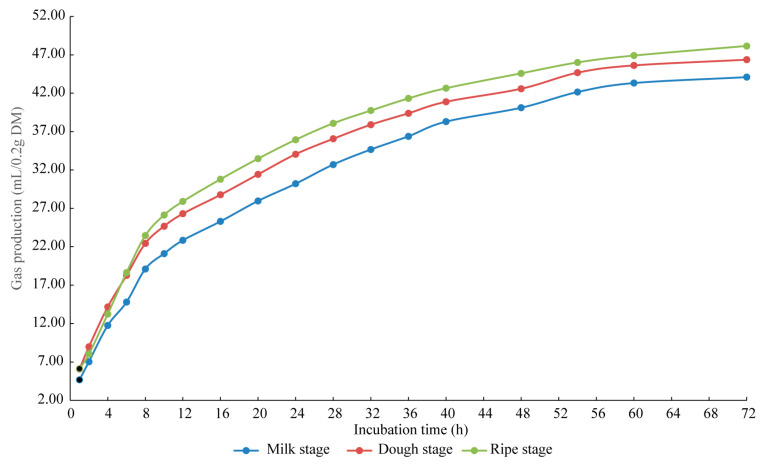
In vitro gas production curves (0–72 h) of sorghum stalk silage with different maturity.

**Figure 3 microorganisms-13-02131-f003:**
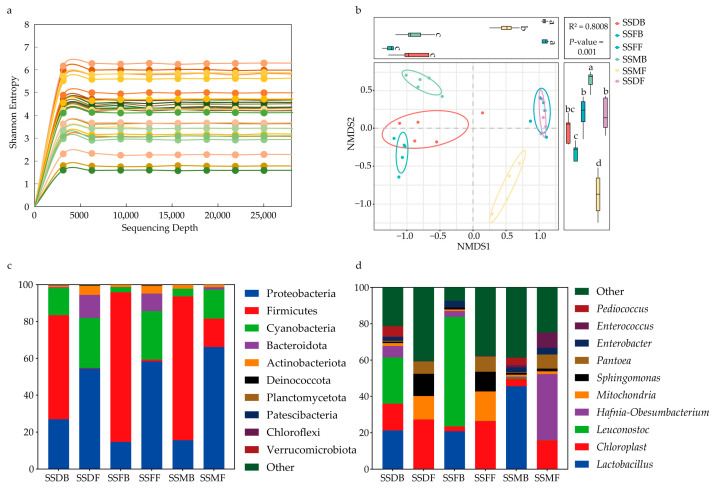
The sorghum stalk bacterial communities as affected by harvesting period and silage. (**a**) Rarefaction analysis of bacteria at the Shannon index. (**b**) Non-metric Multidimensional Scaling (NMDS) based on ASVs of sorghum stalks among different harvesting period before and after ensiling. Different lowercase letters in the figure indicate significant differences among different harvest periods (*p* < 0.05) (**c**) The relative abundances of sorghum stalk bacteria at phylum level before and after ensiling among different harvesting period. (**d**) The relative abundances of sorghum stalks bacteria at genus level before and after ensiling among different harvesting periods. SSMF, sorghum straw before silage at the milk stage; SSDF, sorghum straw before silage at the dough stage; SSFF, sorghum straw before silage at the ripe stage; SSMB, sorghum straw silage at the milk stage; SSDB, sorghum straw silage at the dough stage; SSFB, sorghum straw silage at the ripe stage.

**Figure 4 microorganisms-13-02131-f004:**
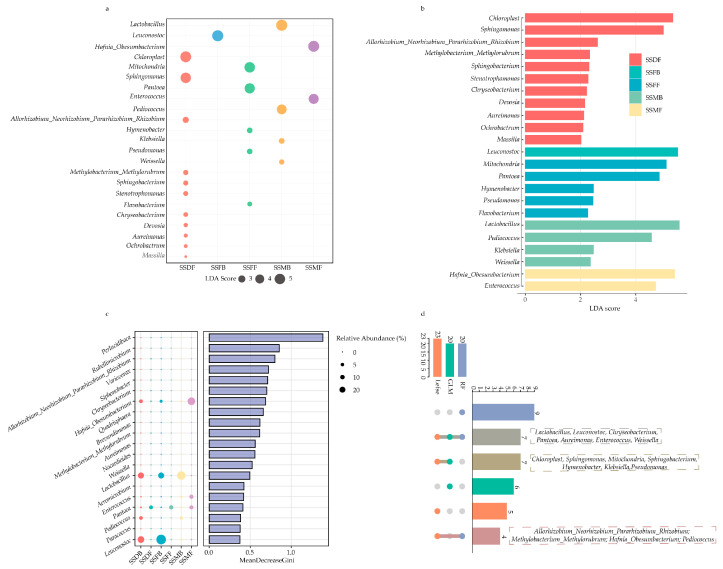
Biomarker bacterial genera distinguishing sorghum stalks across different harvest stages and ensiling treatments. (**a**,**b**) Linear Discriminant Analysis Effect Size (LEfSe) identifying the most differentially abundant genera (LDA score > 2.0) in sorghum stalks across different harvest stages and ensiling treatments. (**c**) Random Forest analysis ranking the top 20 most important sorghum stalks feature genera affected by harvest stages and ensiling treatments based on the Mean Decrease Gini score. (**d**) Summary of the 18 key bacterial genera identified by the intersection of three analytical models (GLM, LEfSe, and Random Forest). SSMF, sorghum straw before silage at the milk stage; SSDF, sorghum straw before silage at the dough stage; SSFF, sorghum straw before silage at the ripe stage; SSMB, sorghum straw silage at the milk stage; SSDB, sorghum straw silage at the dough stage; SSFB, sorghum straw silage at the ripe stage.

**Figure 5 microorganisms-13-02131-f005:**
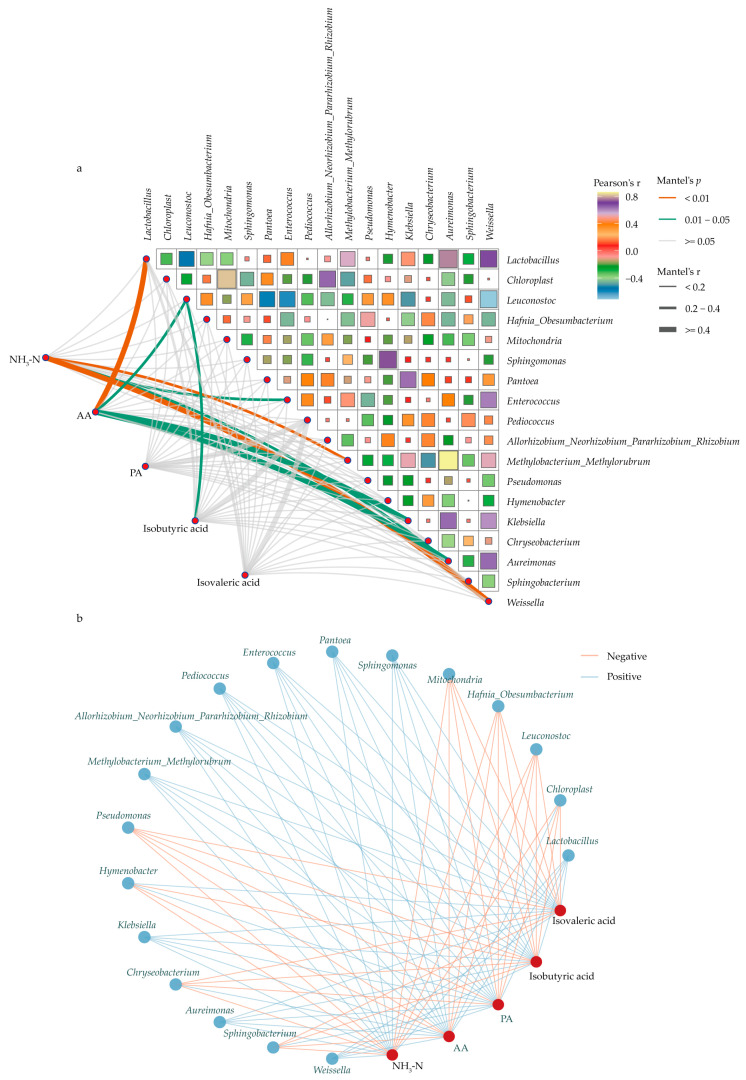
Correlations between key bacteria and in vitro rumen fermentation parameters. (**a**) Relationships between different groups enriched bacteria and fermentation parameters. The correlation was determined by the Mantel test. The edge color and width denote the statistical significance (Mantel *p*) and the correlation coefficient (Mantel r). The color and size of the squares in the upper right corner indicate the strength of the correlation between microorganisms (Pearson’s r). (**b**) Network plots for the key bacteria and fermentation parameters; the lines represent the relationships (blue for positive and orange for negative), and blue nodes represent bacterial genera, while red nodes represent fermentation parameters.

**Table 1 microorganisms-13-02131-t001:** Sorghum stalk silage quality at different harvesting times (DM basis, %) ^1^.

Items	Milk Stage	Dough Stage	Ripe Stage	SEM	*p*-Value
DM	18.96 ^b^	19.37 ^a^	24.21 ^a^	0.691	<0.001
DML	0.99 ^b^	0.85 ^b^	2.73 ^a^	0.260	<0.001
CP	10.47 ^a^	9.03 ^b^	7.81 ^c^	0.293	<0.001
EE	3.89 ^b^	3.63 ^c^	4.18 ^a^	0.071	<0.001
NDF	56.80 ^a^	51.30 ^b^	47.94 ^b^	1.164	<0.001
ADF	31.93 ^a^	29.55 ^a^	27.15 ^b^	2.886	0.024
Ash	10.50	12.29	10.14	0.872	0.591
NFC	18.34 ^b^	23.75 ^ab^	29.95 ^a^	1.670	<0.001
GE (MJ/kg)	20.46 ^a^	19.95 ^b^	19.58 ^c^	0.103	<0.001
RFV	105.51 ^c^	119.57 ^b^	131.60 ^a^	3.452	<0.001
TDN	65.49 ^b^	67.15 ^ab^	68.83 ^a^	0.527	0.023
pH	3.90 ^c^	4.10 ^b^	4.26 ^a^	0.048	<0.001
LA	20.92	19.39	19.00	0.512	0.282
AA	7.90	9.57	6.30	0.615	0.076
LA/AA	2.90	2.08	3.17	0.221	0.104
NH_3_-N (g/kg)	0.54 ^ab^	0.43 ^b^	0.59 ^a^	0.037	0.048

^1^ DM, dry matter; DML, dry matter loss; CP, crude protein; EE, ether extract; NDF, neutral detergent fiber; ADF, acid detergent fiber; NFC, non-fiber carbohydrate; GE, gross energy; RFV, relative feed value; TDN, total digestible nutrients; LA, lactic acid; AA, acetic acid. SEM, standard error of the mean, different lowercase letters in the same row indicate significant difference (*p* < 0.05). *p*-values refer to the main effect of harvest stage. Multiple comparisons were adjusted using Tukey’s test.

**Table 2 microorganisms-13-02131-t002:** Sensory evaluation of sorghum stalk silage in different harvest periods.

Items	Smell	Structure	Color	Total Score	Class
Milk stage	13	3	1	17	Excellent
Dough stage	13	4	2	19	Excellent
Ripe stage	12	3	1	16	Excellent

**Table 3 microorganisms-13-02131-t003:** V-Score of sorghum stalk silage in different harvest periods.

Items	NH_3_-N/TN		AA + PA		Butyric Acid		Score	Class
	Content %	Score	Content %	Score	Content %	Score		
Milk stage	3.50	50	0	10	0	40	100	Good
Dough stage	4.13	50	0	10	0	40	100	Good
Ripe stage	3.71	50	0	10	0	40	100	Good

**Table 4 microorganisms-13-02131-t004:** In vitro fermentation parameters of sorghum stalk silage at different harvest stages ^1^.

Items	Milk Stage	Dough Stage	Ripe Stage	SEM	*p*-Value
GP (mL/0.2 g DM)	44.10 ^b^	46.37 ^ab^	48.16 ^a^	0.611	0.012
b (mL/0.2 g DM)	41.41	41.54	46.40	1.084	0.093
c (mL/h)	0.06	0.07	0.08	0.007	0.067
pH	6.47 ^b^	6.61 ^a^	6.64 ^a^	0.023	<0.001
NH_3_-N (mmol/L)	1.43 ^a^	1.23 ^b^	1.28 ^b^	0.038	<0.001
AA (mmol/L)	64.60 ^a^	54.86 ^ab^	49.07 ^b^	2.462	0.026
PA (mmol/L)	16.51 ^a^	14.83 ^ab^	13.07 ^b^	0.556	0.035
Butyric acid (mmol/L)	5.30	4.70	5.04	0.171	0.371
Isobutyric acid (mmol/L)	0.91 ^a^	0.66 ^b^	0.39 ^c^	0.070	<0.001
Isovaleric acid (mmol/L)	1.76 ^a^	1.16 ^b^	1.07 ^b^	0.091	<0.001
AA/PA (%)	3.91	3.74	3.78	0.106	0.794

^1^ AA, acetic acid; PA, propionic acid; GP, gas production; SEM, standard error of the mean; b, gas production of the slowly degradable fraction mL/g DM; c, rate of gas production of the slowly degradable fraction, %/h. Different lowercase letters in the same row indicate significant difference (*p* < 0.05). *p*-values refer to the main effect of harvest stage. Multiple comparisons were adjusted using Tukey’s test.

**Table 5 microorganisms-13-02131-t005:** Alpha-diversity indexes of bacterial communities in fresh sorghum stalk and its silage at different maturity stages ^1^.

Items	Control			Treatments			SEM	*p*-Value	
	SSMF	SSDF	SSFF	SSMB	SSDB	SSFB		H	S
Shannon index	4.31 ^ab^	5.29 ^a^	5.13 ^a^	3.88 ^b^	4.08 ^b^	2.41 ^c^	0.21	0.055	<0.01
Simpson index	0.87 ^a^	0.89 ^a^	0.88 ^a^	0.83 ^a^	0.83 ^a^	0.59 ^b^	0.02	<0.01	<0.01
Faith PD	12.73 ^bc^	27.83 ^a^	27.55 ^a^	11.89 ^c^	16.94 ^b^	12.62 ^bc^	1.4	<0.01	<0.01
Observed features	160.60 ^b^	410.40 ^a^	396.80 ^a^	137.8 ^b^	206.75 ^b^	125.20 ^b^	23.91	<0.01	<0.01

^1^ SEM, standard error of the mean; SSMF, sorghum straw before silage at the milk stage; SSDF, sorghum straw before silage at the dough stage; SSFF, sorghum straw before silage at the ripe stage; SSMB, sorghum straw silage at the milk stage; SSDB, sorghum straw silage at the dough stage; SSFB, sorghum straw silage at the ripe stage. H, harvest period; S, silage treatment. Different lowercase letters in the same row indicate significant difference (*p* < 0.05).

**Table 6 microorganisms-13-02131-t006:** Relative abundance of top 20 bacterial communities at genus level in sorghum stalk and its silage at different maturity stages (%) ^1^.

Items	Treatments			Control			SEM	*p*-Value	
	SSMB	SSDB	SSFB	SSMF	SSDF	SSFF		H	S
*Chloroplast*	2.69 ^b^	14.71 ^ab^	7.15 ^b^	15.70 ^ab^	41.62 ^a^	26.34 ^a^	9.23	0.102	<0.01
*Lactobacillus*	45.54 ^a^	21.24 ^b^	20.77 ^b^	0.11 ^c^	0.01 ^c^	0.01 ^c^	5.32	<0.01	<0.01
*Leuconostoc*	0.61 ^c^	25.45 ^b^	60.28 ^a^	00.14 ^c^	0.01 ^c^	0.09 ^c^	5.18	<0.01	<0.01
*Hafnia-Obesumbacterium*	0.52 ^b^	6.31 ^b^	3.22 ^b^	36.29 ^a^	0.08 ^b^	0.04 ^b^	2.75	<0.01	<0.01
*Mitochondria*	1.08 ^b^	1.60 ^b^	0.89 ^b^	1.52 ^b^	12.88 ^a^	16.20 ^a^	2.47	<0.01	<0.01
*Lactococcus*	23.62 ^a^	3.26 ^b^	0.03 ^b^	5.75 ^b^	0.01 ^b^	0.01 ^b^	4.65	<0.01	<0.01
*Sphingomonas*	0.96 ^b^	0.77 ^b^	1.16 ^b^	1.49 ^b^	12.24 ^a^	10.81 ^a^	1.93	<0.01	<0.01
*Pantoea*	0.34 ^b^	0.48 ^b^	0.10 ^b^	7.76 ^a^	6.91 ^a^	8.42 ^a^	2.98	0.953	<0.01
*Unclassified_EnterObacteriaceae*	3.40 ^ab^	8.09 ^a^	2.97 ^ab^	6.60 ^a^	0.34 ^b^	0.28 ^b^	3.16	0.201	0.029
*Enterobacter*	2.70 ^a^	2.22 ^a^	3.51 ^a^	3.76 ^a^	0.14 ^a^	0.13 ^a^	2.19	0.267	0.178
*Enterococcus*	1.33 ^b^	0.17 ^b^	0.03 ^b^	8.34 ^a^	0.04 ^b^	0.25 ^b^	1.09	<0.01	<0.01
*Rhizobium*	0.35 ^b^	0.37 ^b^	0.26 ^b^	1.09 ^b^	3.69 ^a^	3.55 ^a^	1.29	0.212	<0.01
*Methylorubrum*	0.91 ^b^	0.41 ^b^	0.49 ^b^	0.47 ^b^	2.83 ^a^	2.10 ^a^	0.46	<0.01	<0.01
*Hymenobacter*	0.00 ^b^	0.00 ^b^	0.01 ^b^	0.05 ^b^	3.44 ^a^	3.07 ^a^	0.42	<0.01	<0.01
*Klebsiella*	3.68 ^a^	1.69 ^b^	0.64 ^b^	0.09 ^b^	0.04 ^b^	0.06 ^b^	0.98	0.068	<0.01
*Aureimonas*	0.94 ^b^	0.36 ^c^	0.43 ^c^	0.17 ^c^	1.63 ^a^	1.11 ^b^	0.28	0.016	<0.01
*Chryseobacterium*	0.01 ^b^	0.03 ^b^	0.02 ^b^	0.81 ^ab^	1.87 ^a^	1.77 ^ab^	0.88	0.057	0.023
*Weissella*	2.83 ^a^	0.65 ^b^	0.02 ^b^	0.26 ^b^	0.01 ^b^	0.01 ^b^	0.40	<0.01	<0.01
*Sphingobacterium*	0.01 ^b^	0.02 ^b^	0.02 ^b^	0.32 ^ab^	1.93 ^a^	1.15 ^ab^	0.82	0.374	0.077
Others	7.03 ^b^	12.18 ^ab^	2.46 ^b^	8.09 ^b^	22.90 ^a^	21.75 ^a^	6.76	0.031	<0.01

^1^ SEM, standard error of the mean; SSMF, sorghum straw before silage at the milk stage; SSDF, sorghum straw before silage at the dough stage; SSFF, sorghum straw before silage at the ripe stage; SSMB, sorghum straw silage at the milk stage; SSDB, sorghum straw silage at the dough stage; SSFB, sorghum straw silage at the ripe stage. H, harvest period; S, silage treatment. Different lowercase letters in the same row indicate significant difference (*p* < 0.05).

## Data Availability

Raw sequences of the 16S rRNA gene region in FASTQ file format for all samples have been deposited in the NCBI SRA repository under the BioProject accession number PRJNA1304916.

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
