# Peer review of "Harvest Stage Dictates the Nutritive Value of Sorghum Stalk Silage by Shaping Its Fermentation Profile and Microbial Composition"

_microorganisms, 2025, doi:10.3390/microorganisms13092131_

Round 1
Reviewer 1 Report
Comments and Suggestions for Authors
The article was thoroughly reviewed. The following corrections were made:
In lines 43-44: In China, sorghum cultivation spanned 729,133 hectares in 2022, yielding 3.32 million tons and generating about 4.33 million tons of stalk [8]. A review of the literature on this grass revealed no such information. The literature needs correction.
In ilne 45: In addition, sorghum stalk possesses high feeding value, being rich in fiber, protein, sugar, and essential amino acids [9]. Literature number 9 is about buckwheat straw. This needs correction.
In lines 48-50: However, the application of sorghum straw in ruminant production is still limited since it is stable in structure, hard in texture, and rich in lignin, cellulose, and hemicellulose [11]. A review of citation number 11 reveals no such information. Only one sentence mentions that sorghum straw is ineffective as a ruminant feed. The article is a study on the effects of enzyme addition to sorghum silage. The sentence needs to be either corrected or supported with appropriate literature.
In line 56. Citation number 13 is not relevant to this study. The effects of dietary fiber were investigated in pigs.
The purpose of the study in introduction is missing. Therefore, the purpose of this study……………………………….
Line 88. Should be Menke and Steingass [19]
Lines 103-108-110-113 should be AOAC [20]
Line 112. (1991) should be deleted
Line 250. Explanations for b and c should be provided in Table 4.
Line 563. 1 in the title should be deleted.
Lines 464-478-492-501-513-519-529- All author names should be given.
Author Response
Dear Editors and reviewer
We would like to express sincere appreciation to you and the anonymous reviewers who had given these constructive comments and suggestions. These comments and suggestions are all valuable and very helpful for revising and improving our manuscript entitled “Harvest Stage Dictates the Nutritive Value of Sorghum Silage by Shaping its Fermentation Profile and Microbial Composition”, as well as the important guiding significance for our researches.
We have revised the manuscript according to your kind advices, and all changes have been marked in red in the revised manuscript. Thank you very much for all your help and looking forward to hearing from you soon.
Point 1: In lines 43-44: In China, sorghum cultivation spanned 729,133 hectares in 2022, yielding 3.32 million tons and generating about 4.33 million tons of stalk [8]. A review of the literature on this grass revealed no such information. The literature needs correction.
Response: We sincerely thank you for pointing out this mistake. After carefully re-checking the cited reference, we confirmed that it did not contain the reported data. Therefore, we have corrected this sentence by using the official statistics from FAOSTAT. The incorrect reference has been removed, and the FAOSTAT source has been cited instead. Please see the Lines 45-47 of the revised manuscript.
Point 2: In line 45: In addition, sorghum stalk possesses high feeding value, being rich in fiber, protein, sugar, and essential amino acids [9]. Literature number 9 is about buckwheat straw. This needs correction.
Response: We sincerely thank you for this valuable suggestion regarding the literature. We agree that the previous reference [9] was inappropriate and have replaced it with a more relevant source (Etuk et al.). Meanwhile, the reference list has been updated accordingly. Please see Line 49 of the revised manuscript.
Etuk, E.B.; Ifeduba, A.V.; Okata, U.E.; Chiaka, I.; Charles, O.I.; Okeudo, N.J.; Esonu, B.O.; Udedibie, A.B.I.; Moreki, J.C. Nutrient Composition and Feeding Value of Sorghum for Livestock and Poultry: A Review. Anim. Sci. Adv. 2012, 2, 510–524.
Point 3: In lines 48-50: However, the application of sorghum straw in ruminant production is still limited since it is stable in structure, hard in texture, and rich in lignin, cellulose, and hemicellulose [11]. A review of citation number 11 reveals no such information. Only one sentence mentions that sorghum straw is ineffective as a ruminant feed. The article is a study on the effects of enzyme addition to sorghum silage. The sentence needs to be either corrected or supported with appropriate literature.
Response: We sincerely thank you for your helpful comment. We agree that the previous reference [11] was inappropriate, and we have replaced it with a more relevant source (Bakari et.al). Meanwhile, the reference list has been updated accordingly. Please see Line 53 of the revised manuscript.
Bakari, H.; Djomdi; Ruben, Z.F.; Roger, D.D.; Cedric, D.; Guillaume, P.; Pascal, D.; Philippe, M.; Gwendoline, C. Sorghum (Sorghum Bicolor L. Moench) and Its Main Parts (by-Products) as Promising Sustainable Sources of Val-ue-Added Ingredients. Waste Biomass Valor 2023, 14, 1023–1044, doi:10.1007/s12649-022-01992-7.
Point 4: In line 56. Citation number 13 is not relevant to this study. The effects of dietary fiber were investigated in pigs.
Response: We sincerely thank you for pointing out this issue. We agree that the previous reference [13] was not relevant and have replaced it with a more appropriate source (Ambye-Jensen et al.). The reference list has been updated accordingly. Please see Line 58 of the revised manuscript.
Ambye-Jensen, M.; Johansen, K.S.; Didion, T.; Kádár, Z.; Meyer, A.S. Ensiling and Hydrothermal Pretreatment of Grass: Consequences for Enzymatic Biomass Conversion and Total Monosaccharide Yields. Biotechnol Biofuels 2014, 7, 95. https://doi.org/10.1186/1754-6834-7-95.
Point 5: The purpose of the study in introduction is missing. Therefore, the purpose of this study……………………………….
Response: Thank you for your valuable suggestion. We have added a clear statement of the study’s purpose at the end of the Introduction section. Please see Lines 71–75 of the revised manuscript.
Point 6: Line 88. Should be Menke and Steingass [19]
Response: We thank you for pointing out this mistake. We have corrected the citation to “Menke and Steingass [19]” in Line 103 of the revised manuscript.
Point 7: Lines 103-108-110-113 should be AOAC [20]
Response: Thank you for your helpful suggestion. We have carefully revised the manuscript and corrected the citations in Lines 119, 124, 126, and 129 to “AOAC [19]” as recommended.
Point 8: Line 112. (1991) should be deleted
Response: Thank you for your helpful suggestion. We have deleted “(1991)” from Line 128 in the revised manuscript.
Point 9: Line 250. Explanations for b and c should be provided in Table 4.
Response: We sincerely thank you for your valuable suggestion. We have revised Table 4 and added clear explanations for the parameters b (gas production of the slowly degradable feed fraction, mL/g DM) and c (gas production rate of the slowly degradable feed fraction, %/h) in the table footnote. Please see Table 4 in the revised manuscript.
Point 10: Line 563. 1 in the title should be deleted.
Response: We thank the reviewer for this observation. We would like to clarify that this is not the number “1,” but rather the author’s name “Tlahig” We have carefully checked the reference and ensured that it is now presented correctly in the revised manuscript.
Point 11: Lines 464-478-492-501-513-519-529- All author names should be given.
Response: Thank you for your helpful suggestion. We have revised the reference list and provided the full author names for all the cited references at Lines 509, 537, 546, 565 and 584 of the revised manuscript.
Reviewer 2 Report
Comments and Suggestions for Authors
This manuscript studied the impact of harvest maturity on sorghum silage nutritive value, fermentation quality, and bacterial community composition. Sorghum silage was collected at the three age phases (milk, dough, and ripe). The results showed that ripe sorghum had higher quality and boosted nutritional value and stability. The study addresses an important topic in forage microbiology and ruminant nutrition; however, several issues require revision and clarification before the manuscript can be considered for publication:
- There are some typos, such as “and ensiled for 60 d with at 15–25°C” (Line 80), “was fist applied to assess” line 307), "GML" model in the results (lines 307, 334) should likely be "GLM" (General Linear Model), "isobutylate" should be "isobutyrate," and some axis labels in figures are misspelled ("Lactonotace" for Leuconostoc in Fig 5). These should be corrected in proofreading.
- Were any negative controls, such as extraction blanks and PCR blanks, sequenced alongside the samples? If so, were their profiles subtracted from the data? This is a critical step to ensure that low-abundance taxa are not kit or laboratory contaminants.
- Multiple ANOVA comparisons are presented without a clear adjustment for false discovery rate (FDR). This increases the risk of type I errors.
- Some statistical information is missing in table 1 and 4.
- Names of all bacterial taxa are printed in italics and should be italicized in the manuscript, so please italicize the names in figures 3, 4, and 5 and in line 439.
- Figure 1, ensure the figure caption clearly defines what is being measured on the y-axis and x-axis (Time, h).
- Figure 2, its quality is bad, and it is not easy to see what is measured on the y-axis and x-axis.
- Figure 4, it is difficult to interpret due to the high number of taxa. It would be beneficial to focus on the top 10 genera. You can provide the full figure as a supplement.
- Figure 5, the Mantel test is a good choice for correlating community data with environmental variables, but it is currently confusing. For example, in the network plot (Fig. 5B), the labels are cut off, and the meaning of the node shapes and sizes is not explained in the caption. This figure needs a thorough legend and editing for clarity.
Author Response
Dear Editors and reviewer
We would like to express sincere appreciation to you and the anonymous reviewers who had given these constructive comments and suggestions. These comments and suggestions are all valuable and very helpful for revising and improving our manuscript entitled “Harvest Stage Dictates the Nutritive Value of Sorghum Silage by Shaping its Fermentation Profile and Microbial Composition”, as well as the important guiding significance for our researches.
We have revised the manuscript according to your kind advices, and all changes have been marked in red in the revised manuscript. Thank you very much for all your help and looking forward to hearing from you soon.
Responses to Reviewer 2:
Point 1: There are some typos, such as “and ensiled for 60 d with at 15–25°C” (Line 80), “was fist applied to assess” line 307), "GML" model in the results (lines 307, 334) should likely be "GLM" (General Linear Model), "isobutylate" should be "isobutyrate," and some axis labels in figures are misspelled ("Lactonotace" for Leuconostoc in Fig 5). These should be corrected in proofreading.
Response: We appreciate you for your careful observation. The following corrections in the revised manuscript have been made:
Line 90: revised to “and ensiled for 60 d at 15–25°C”.
Line 335: corrected “fist” to “first”.
Lines 335 and 362: corrected “GML” to “GLM”.
For Figure 5, we carefully rechecked the spelling of all axis labels and confirmed that the genus name “Leuconostoc” was already spelled correctly.
In addition, we have thoroughly proofread the entire manuscript and corrected other similar typographical and spelling issues to ensure consistency and accuracy throughout.
Point 2: Were any negative controls, such as extraction blanks and PCR blanks, sequenced alongside the samples? If so, were their profiles subtracted from the data? This is a critical step to ensure that low-abundance taxa are not kit or laboratory contaminants.
Response: Thank you for your important comment. Negative controls (extraction blanks and PCR blanks) were indeed processed and sequenced alongside the experimental samples. These controls yielded negligible read counts, and no consistent ASVs were detected. We have added this information in the Materials and Methods. Please see the Lines 167-191 in the revised manuscript.
Point 3: Multiple ANOVA comparisons are presented without a clear adjustment for false discovery rate (FDR). This increases the risk of type I errors.
Response: Thank you for your valuable comment. In this study, one-way ANOVA was followed by Tukey’s test for multiple comparisons. Tukey’s test is a widely accepted post-hoc method that controls the family-wise error rate (FWER) and is widely applied in animal nutrition and silage studies to limit type I errors.
Point 4: Some statistical information is missing in table 1 and 4.
Response: Thank you for pointing this question. We have revised Table 1 and Table 4 to include the missing statistical details. Please see the Lines 241-242 and Lines 279-280 in the revised manuscript.
Point 5: Names of all bacterial taxa are printed in italics and should be italicized in the manuscript, so please italicize the names in figures 3, 4, and 5 and in line 439.
Response: We appreciate your attention to the formatting of microbial taxa. According to the journal’s published articles and standard taxonomic conventions, genus and species names are italicized, whereas phylum are written in regular font. Following this principle, we have carefully revised the manuscript: all genus-level taxa are now italicized (including in Figures 3, 4, and 5, and in line 467), while phylum-level taxa remain in regular font. This ensures consistency with both journal style and nomenclatural standards.
Point 6: Figure 1, ensure the figure caption clearly defines what is being measured on the y-axis and x-axis (Time, h).
Response: Thank you for your helpful suggestion. In the revised Figure 1, the y-axis label is presented concisely as “Aerobic stability (h).” To avoid ambiguity, the figure caption now provides the full definition: aerobic stability was defined as the time required for the temperature at the geometric center of the silage to rise 2 °C above ambient temperature.
Point 7: Figure 2, its quality is bad, and it is not easy to see what is measured on the y-axis and x-axis.
Response: Thank you for your valuable comment. Figure 2 has been redrawn and replaced with a high-resolution version to improve clarity. In addition, we have revised the figure caption to clearly define the axes: the y-axis represents cumulative gas production (mL/0.2 g DM), and the x-axis represents incubation time (h).
Point 8: Figure 4, it is difficult to interpret due to the high number of taxa. It would be beneficial to focus on the top 10 genera. You can provide the full figure as a supplement.
Response: Thank you for your constructive suggestion. We agree that simplifying the figure can improve readability. However, in this study, key microorganisms were identified by combining three complementary approaches (GLM, Random Forest, and LEfSe), and then taking the intersection of their results. To ensure a comprehensive representation of these shared key taxa, we focused on the top 20 genera rather than the top 10. Restricting the analysis to only the top 10 genera would markedly reduce the overlap across the three methods and could omit biologically relevant taxa that contribute to the core microbial community structure. Therefore, we have retained the top 20 genera in Figure 4 for scientific completeness, while improving figure clarity (larger fonts, simplified layout).
Point 9: Figure 5, the Mantel test is a good choice for correlating community data with environmental variables, but it is currently confusing. For example, in the network plot (Fig. 5B), the labels are cut off, and the meaning of the node shapes and sizes is not explained in the caption. This figure needs a thorough legend and editing for clarity.
Response: Thank you for your constructive suggestion. In the revised manuscript, Figure 5 has been improved for clarity. The labels of long genus names (e.g., Allorhizobium_Neorhizobium_Pararhizobium_Rhizobium, Methylobacterium_Methylorubrum) have been reformatted and the figure margins adjusted to ensure that all names are fully visible without being cut off. In addition, the figure caption has been expanded to provide a thorough explanation of the figure elements. Please see the Lines 379-383 in the revised manuscript.
Reviewer 3 Report
Comments and Suggestions for Authors
Dear Authors,
The article presents consistent and well-analyzed results, the discussion is adequate, and the conclusion addresses the stated objective. Some suggestions are recommended to strengthen the practical context and balance the interpretation of the benefits and disadvantages of harvesting at the ripening stage.
Abstract:
I suggest that the authors add the experimental design, number of replications, and what was evaluated prior to presenting the results.
Keywords:
Keywords should avoid terms already present in the article title.
Introduction:
I suggest including the research objective at the end of the introduction.
Materials and Methods:
Item 2.1
If animals were manipulated in this research (rumen fluid collection), please include the protocol number from the Institutional Animal Care and Use Committee (IACUC).
I recommend adding the geographic coordinates of the experimental site as well as climatic data (precipitation, temperature, humidity, etc.).
Specify the sorghum genotype used.
Clarify the exact dimensions of the experimental 5 L silo.
Item 2.2
What diet were the six beef cattle fed? Was there an adaptation period before rumen fluid collection?
Item 2.6
The calculations for TDN, non-fiber carbohydrates, and relative feed value are described in the methods, but a supporting citation is required.
Which normality test was applied in the statistical analysis?
Results:
Table 1: It should be highlighted that the TDN value at the ripe stage was not statistically higher than at the dough stage, and the ADF value at the milk stage was not higher than at the dough stage.
The increase in DM, RFV, and TDN at the ripe stage is clearly demonstrated, but the reduction in CP deserves more discussion in terms of practical feeding strategies (protein supplementation may be necessary).
Discussion:
The authors should emphasize the trade-off between lower CP and higher digestibility at the ripe stage. A more explicit comparison with studies on corn silage would reinforce the practical implications of sorghum silage as an alternative.
The discussion sometimes overstates the superiority of the ripe stage without sufficiently balancing the trade-off between higher digestibility and lower protein content.
The correlation analysis (Figure 5) is promising, but the biological significance of some associations (e.g., AA with Lactobacillus and Klebsiella) should be elaborated further.
Conclusion:
The conclusion directly addresses the objective but is too categorical. A more nuanced statement acknowledging that ripe-stage silage may require dietary adjustments (protein supplementation) would strengthen the practical value of the findings.
References:
I noticed that one reference appears duplicated (Refs. 34 and 37). Please verify and remove the duplication.
I recommend standardizing the reference style according to the journal’s guidelines, as some journal names are written in full while others are abbreviated.
Author Response
Dear Editors and reviewer
We would like to express sincere appreciation to you and the anonymous reviewers who had given these constructive comments and suggestions. These comments and suggestions are all valuable and very helpful for revising and improving our manuscript entitled “Harvest Stage Dictates the Nutritive Value of Sorghum Silage by Shaping its Fermentation Profile and Microbial Composition”, as well as the important guiding significance for our researches.
We have revised the manuscript according to your kind advices, and all changes have been marked in red in the revised manuscript. Thank you very much for all your help and looking forward to hearing from you soon.
Point 1: Abstract: I suggest that the authors add the experimental design, number of replications, and what was evaluated prior to presenting the results.
Response: Thank you for your helpful suggestion. We have added details of the experimental design, including the harvest stages, number of replications, and evaluation parameters, before presenting the results. Please see Lines 13 - 18 of the revised manuscript.
Point 2: Keywords should avoid terms already present in the article title.
Response: Thank you for your helpful suggestion. In the revised manuscript, we have modified the keywords to reduce redundancy with the title while maintaining scientific accuracy and indexing value. The revised keywords are: sorghum stalk; forage quality; maturity stages; In vitro rumen fermentation; microbial community.
Point 3: Introduction: I suggest including the research objective at the end of the introduction.
Response: Thank you for your valuable suggestion. We have added a clear statement of the study’s purpose at the end of the Introduction section. Please see Lines 71–75 of the revised manuscript.
Point 4: Item 2.1 If animals were manipulated in this research (rumen fluid collection), please include the protocol number from the Institutional Animal Care and Use Committee (IACUC). I recommend adding the geographic coordinates of the experimental site as well as climatic data (precipitation, temperature, humidity, etc.). Specify the sorghum genotype used. Clarify the exact dimensions of the experimental 5 L silo.
Response: Thank you for your helpful suggestion. We have added the IACUC protocol number, geographic coordinates and climatic data of the experimental site, specified the sorghum genotype used, and clarified the exact dimensions of the 5 L silos. Please see Lines 79 - 83 Line 88 and Lines 93-94 of the revised manuscript.
Point 5: Item 2.2 What diet were the six beef cattle fed? Was there an adaptation period before rumen fluid collection?
Response: Thank you for pointing this out. We have added details of the diet and feeding management of the six beef cattle, and specified that an adaptation period was provided before rumen fluid collection. Please see Lines 96 - 100 of the revised manuscript.
Point 6: Item 2.6 The calculations for TDN, non-fiber carbohydrates, and relative feed value are described in the methods, but a supporting citation is required.
Which normality test was applied in the statistical analysis?
Response: Thank you for this valuable suggestion. Supporting citations have been added for the calculations of TDN, non-fiber carbohydrates, and relative feed value. The Shapiro - Wilk test has been specified as the method used to verify data normality. Please see Lines 195–199 Line 216 and of the revised manuscript.
Point 7: Table 1: It should be highlighted that the TDN value at the ripe stage was not statistically higher than at the dough stage, and the ADF value at the milk stage was not higher than at the dough stage. The increase in DM, RFV, and TDN at the ripe stage is clearly demonstrated, but the reduction in CP deserves more discussion in terms of practical feeding strategies (protein supplementation may be necessary).
Response: Thank you for your helpful suggestion. The results section has been revised to clearly state that the TDN value at the ripe stage was not significantly higher than at the dough stage, and that the ADF value at the milk stage did not differ significantly from the dough stage but both were higher than at the ripe stage (Lines 232 – 236 of revise manuscript). In addition, the Discussion has been supplemented with practical implications of the reduced CP content at the ripe stage, highlighting that protein supplementation may be required when using ripe-stage sorghum silage in ruminant diets (Lines 406 - 411 of revise manuscript).
Point 8: The authors should emphasize the trade-off between lower CP and higher digestibility at the ripe stage. A more explicit comparison with studies on corn silage would reinforce the practical implications of sorghum silage as an alternative. The discussion sometimes overstates the superiority of the ripe stage without sufficiently balancing the trade-off between higher digestibility and lower protein content.
Response: Thank you for your constructive comment. In the revised manuscript, the Discussion section has been expanded to address the practical implications of the reduced CP concentration at the ripe stage. We now note that lower CP levels could negatively affect nutrient digestibility, and protein supplementation may therefore be required in practical rations. At the same time, it is emphasized that although the CP content declined at the ripe stage, it remained higher than values typically reported for corn stover silage, indicating that ripe-stage sorghum silage still possesses a relatively higher protein value compared to conventional corn stover silage (Lines 406 - 411 of revise manuscript).
Point 9: The correlation analysis (Figure 5) is promising, but the biological significance of some associations (e.g., AA with Lactobacillus and Klebsiella) should be elaborated further.
Response: Thank you for your insightful suggestion. The Discussion section has been revised to elaborate on the biological meaning of these associations. Specifically, we clarified that the positive association between Lactobacillus and AA indicates that its enrichment at the ripe stage was linked with efficient lactic acid fermentation and superior overall silage quality. In addition, we explained that the positive association between Klebsiella and AA suggests that part of the acetate in the milk stage silage likely originated from Klebsiella-driven fermentation, which associated with ideal fermentation that explained the inferior preservation efficiency at milk stage. Please see Lines 460-462 and Lines 468-472 of the revised manuscript.
Point 10: The conclusion directly addresses the objective but is too categorical. A more nuanced statement acknowledging that ripe-stage silage may require dietary adjustments (protein supplementation) would strengthen the practical value of the findings.
Response: Thank you for your constructive suggestion. We have revised the Conclusion to provide a more balanced statement. In addition to identifying the ripe stage as the optimal harvest time, we now acknowledge that the lower crude protein concentration at this stage may necessitate dietary protein supplementation to ensure a balanced ration and sustain animal performance. Please see Lines 482-485 of the revised manuscript.
Point 11: I noticed that one reference appears duplicated (Refs. 34 and 37). Please verify and remove the duplication. I recommend standardizing the reference style according to the journal’s guidelines, as some journal names are written in full while others are abbreviated.
Response: Thank you for your helpful suggestion. The duplicated reference has been removed, and the entire reference list has been standardized according to the journal’s formatting requirements. Please see the revised reference section.